# Extreme Value Statistics of Community Detection in Complex Networks with Reduced Network Extremal Ensemble Learning (RenEEL)

**DOI:** 10.3390/e27060628

**Published:** 2025-06-13

**Authors:** Tania Ghosh, Royce K. P. Zia, Kevin E. Bassler

**Affiliations:** 1Department of Physics, University of Houston, Houston, TX 77204, USA; tghosh3@uh.edu (T.G.); rkpzia@vt.edu (R.K.P.Z.); 2Texas Center for Superconductivity, University of Houston, Houston, TX 77204, USA; 3Center for Soft Matter and Biological Physics, Department of Physics, Virginia Tech, Blacksburg, VA 24061, USA; 4Department of Mathematics, University of Houston, Houston, TX 77204, USA

**Keywords:** community detection, complex networks, modularity, extreme value statistics

## Abstract

Arguably, the most fundamental problem in Network Science is finding structure within a complex network. Often, this is achieved by partitioning the network’s nodes into communities in a way that maximizes an objective function. However, finding the maximizing partition is generally a computationally difficult NP-complete problem. Recently, a machine learning algorithmic scheme was introduced that uses information within a set of partitions to find a new partition that better maximizes an objective function. The scheme, known as RenEEL, uses Extremal Ensemble Learning. Starting with an ensemble of *K* partitions, it updates the ensemble by considering replacing its worst member with the best of *L* partitions found by analyzing a reduced network formed by collapsing nodes, which all the ensemble partitions agree should be grouped together, into super-nodes. The updating continues until consensus is achieved within the ensemble about what the best partition is. The original *K* ensemble partitions and each of the *L* partitions used for an update are found using a simple “base” partitioning algorithm. We perform an empirical study of how the effectiveness of RenEEL depends on the values of *K* and *L* and relate the results to the extreme value statistics of record-breaking. We find that increasing *K* is generally more effective than increasing *L* for finding the best partition.

## 1. Introduction

Finding the community structure within a complex network that relates to its function or behavior is a fundamental challenge in Network Science [1,2,3,4,5]. It is a highly non-trivial problem, as even defining what one means by “structure” must be specified [6]. A variety of approaches can be used to partition the nodes of a network into communities [7,8]. Each approach will divide the nodes according to a different definition of the structure and, in general, will find a different partition [9,10]. Often, the goal is to find a partition that maximizes an objective function. However, finding such a partition can be a computationally difficult NP-complete problem [11,12]. Finding a guaranteed exact solution for a large network is therefore generally infeasible. Thus, for practical applications, it is important to have an approximate algorithm that has polynomial-time complexity, i.e., is fast, and that finds near-exact best solutions for large networks, i.e., is accurate.

Recently, an algorithmic scheme has been introduced that uses information in an ensemble of partitions to produce a better, more accurate partition when seeking to maximize an objective function. The scheme, Reduced Network Extremal Ensemble Learning (RenEEL) [13], uses a machine learning paradigm for graph partitioning, Extremal Ensemble Learning (EEL). EEL begins with an ensemble $P_{K}$ of k≤K unique partitions. It then iteratively updates the ensemble using extremal criteria until consensus is reached within the ensemble about what the “best” partition is, i.e., the one with the largest value of the objective function. Each update considers adding a new partition *P* to $P_{K}$ as follows: If $P\in P_{K}$, then the “worst” partition in $P_{K}$, $P_{\mathrm{worst}}$, is removed from $P_{K}$, reducing the size of $P_{K}$ by one k→k−1, and the update is complete. If $P\notin P_{K}$ and *P* is worse than $P_{\mathrm{worst}}$, then again $P_{\mathrm{worst}}$ is removed from $P_{K}$, reducing the size of $P_{K}$ by one k→k−1, and the update is complete. If $P\notin P_{K}$ and *P* is better than $P_{\mathrm{worst}}$, then if k=K, $P_{\mathrm{worst}}$ is replaced by *P* in $P_{K}$ and the update is complete, or if k<K, *P* is added to $P_{K}$, increasing the size of $P_{K}$ by one k→k+1, and the update is complete. Iterative updates continue until k=1. The remaining partition is the consensus choice for the best partition.

To ensure fast convergence to a consensus choice in EEL updates, RenEEL conserves the consensus within $P_{K}$ that exists up to that point each time it finds a new partition to be used in an update. It achieves this by partitioning a *reduced network* rather than the original network. The reduced network is constructed by collapsing the nodes that every partition in $P_{K}$ agrees should be in the same community into “super” nodes. Reduced networks are smaller than the original network and can be analyzed faster, focusing effort only on improving partitioning where there is disagreement within $P_{K}$. The consensus within $P_{K}$ increases monotonically, and the size of the reduced networks decreases monotonically as the EEL updates are made. RenEEL creates an ensemble $P_{L}$ consisting of *L* partitions found by analyzing the reduced network to decide the partition to use in each EEL update. The best partition in $P_{L}$ is then used in the update.

There is wide flexibility within the RenEEL scheme. A *base algorithm* is used to find the partitions that initially form the $P_{K}$ ensemble and those that form each $P_{L}$ ensemble. The base algorithm can be any algorithm that finds a partition that maximizes an objective function. Multiple base algorithms can even be used. There is also freedom in choosing the values of *K* and *L*, which represent the maximum size of $P_{K}$ and the size of each $P_{L}$, respectively. The best base algorithm choice and values of *K* and *L* depend on the network being analyzed, desired accuracy, and available computational resources.

This paper investigates the effect of varying *K* and *L* on the performance of RenEEL. Larger values of *K* and *L* will typically lead to a final, consensus best partition with larger values of the objective function [13]. But how does the value of the objective function of the consensus partition reached typically depend on *K* and *L*? How quickly is the value found expected to approach its true maximum value? Given only limited computational resources, is it better to increase *K* or *L*? We empirically study these questions when seeking the partition of three well-known real-world networks that maximizes the objective function Modularity.

## 2. Results

A commonly used approach to find structure in a complex network is to partition the nodes into communities that are more densely connected than expected in a random network. In this approach, the community structure corresponds to the partition that maximizes an objective function called *Modularity* [2,14]. For a given nodal partition $C\equiv \{c_{1},c_{2},\ldots \}$, Modularity *q* is defined as (1)$$q\left(C\right)=\frac{1}{2m}\sum_{\langle ij\rangle }\left[A_{ij}-\frac{k_{i}k_{j}}{2m}\right]\delta_{c_{i}c_{j}},$$
where the sum is over all pairs of nodes 〈ij〉, $c_{i}$ is the community of the *i*th node, and *m* is the total number of links present in the network. $k_{i}$ and $A_{ij}$ are, respectively, the degree of the *i*th node and the ijth element of the adjacency matrix. Thus, Modularity is the difference between the fraction of links inside the partition’s communities, the first term in Equation (Equation 1), and what the expected fraction would be if all links of the network were randomly placed, the second term in Equation (Equation 1). The task is to find the partition *C* that maximizes *q*. We denote the maximum value of *q* as *Q*, the value of which is called “the Modularity” of the network.

A number of algorithms with polynomial-time complexity have been developed to find a partition that maximizes Modularity. They range from very fast but not-so-accurate algorithms, such as the Louvain method [15] or randomized greedy agglomerative hierarchical clustering [16], to more accurate but slower algorithms [17], such as one that combines both agglomeration and division steps [18,19]. The accuracy of all of the algorithms tends to decrease as the size of the network increases.

All of the fast Modularity-maximizing algorithms are stochastic because at intermediate steps of their execution, there are seemingly equally good choices to make that are randomly made. In the end, those choices can be consequential because different runs of an algorithm with different sets of random intermediate choices can result in different solutions. Because of this, multiple runs of an algorithm are often made, say 100, to analyze a network, producing an ensemble of approximate partitions. The partition in the ensemble with the largest Modularity is then taken as the network’s community structure, while all other partitions in the ensemble are discarded. RenEEL instead uses the information within the ensemble to find a more accurate partition.

Here we use a RenEEL algorithm that has a randomized greedy base algorithm [16] to find the community structure by maximizing Modularity in real-world networks A, B, and C. Network A is the As-22july06 Network [20]. It is a snapshot in time of the structure of the Internet at the level of autonomous systems. It has 22,963 nodes, which represent autonomous systems, and 48,436 links of data connection. Network B is the PGP Network [20]. It is a snapshot in time of the giant component of the Pretty-Good-Privacy (PGP) algorithm user network. It has 10,680 nodes, which are the users of the PGP algorithm, and 24,316 links indicating the interactions among them. Lastly, Network C is the Astro-ph network. It is a coauthorship network of scientists in Astrophysics consisting of 16,706 nodes representing scientists and 12,1251 links representing coauthorship in preprints on the Astrophysics arXiv database [21].

For each of the three networks, 300 different runs of RenEEL were made for independent values of *K* and *L* of 10, 20, 40, 80, 160, and 320, respectively. The compute time required to find the consensus partition was measured for each run. The mean and standard errors of the compute times for the runs at a given value of *K* and *L* were then calculated. The full results are listed in Table A1, Table A4 and Table A7 in Appendix A. For a fixed value of *L* or *K*, we find that the mean compute times 〈t〉 increase asymptotically as a power of the other ensemble size,(2)$$\langle t\rangle {\left.∼K^{\alpha_{K}}\right|}_{L\;\mathrm{fixed}}\;\mathrm{and}\;\langle t\rangle {\left.∼L^{\alpha_{L}}\right|}_{K\;\mathrm{fixed}}.$$

For example, Figure 1 shows this power-law behavior for Network A when *L* and *K* have fixed values of 80.

Two-parameter, nonlinear least-squares fits to data for ensemble sizes greater than 10 were then used to determine the proportionality constant and α. Table 1a,b show the values for $\alpha_{K}$ and $\alpha_{L}$, respectively, that result from fits at different fixed values of *L* and *K* for each of the three networks. All statistical errors reported in this paper are ±2σ.

The values of the exponents $\alpha_{K}\left(L\right)$ and $\alpha_{L}\left(K\right)$ weakly vary with the value of *L* and *K*, respectively. The standard errors of $\alpha_{K}\left(L\right)$ and $\alpha_{L}\left(K\right)$ tend to remain consistent between smaller and larger ensemble sizes. The distribution of computed time, as depicted in Figure A1 for Network A, does not follow a normal distribution. Consequently, increasing the ensemble size does not lead to a decrease in the standard error. For each network, however, the value of $\alpha_{K}$ is significantly larger than $\alpha_{L}$. Thus, the expected compute time increases faster with *K* than *L*. Given that larger values of *K* and *L* typically lead to a better result, i.e., a consensus partition with a larger *Q*, one might naively conclude that it is better to increase *L* rather than *K*. But, to determine if that conclusion is, in fact, correct, the way that *Q* increases with *K* and *L* must be taken into account.

To this end, we begin by noting that for any finite-size network, there is only a finite number of possible partitions. Many modularity-maximizing algorithms will consistently find the actual best partition for very small networks. As the network size and number of possible partitions grow, the task becomes harder; algorithms start to fail to find the exact solution and only provide estimates of the actual, or exact, best partition. RenEEL appears to perform very well at finding the actual best partition of networks of sizes of up to a few thousand nodes [13]. Still, even RenEEL can only find estimates of the exact best partition of larger networks, such as the three we analyze in this paper. As values of *K* and *L* increase, the estimates improve, and the value of *Q* of the consensus partition approaches $Q_{\max}$, the Modularity of the exact best partition. To explore how the values of *Q* of RenEEL’s consensus partitions approach $Q_{\max}$ as a function of *K* and *L*, the mean and standard errors of *Q* found in the runs that were made on each network were calculated as a function of *K* and *L*. The results are listed in Table A2, Table A3, Table A5, Table A6, Table A8 and Table A9 in Appendix A. For a fixed value of *L* or *K*, we find that *Q* approaches a maximum value, $Q_{\max}$, as a power-law of the other ensemble size,(3)Q≈Qmax−AKK−γKL fixedand Q≈Qmax−ALL−γLK fixed,
where the *A*s are constants. Figure 2 shows this behavior for Network A when *L* and *K* have fixed values of 80. The exact value of $Q_{\max}$ is unknown for Networks A, B, and C. Three-parameter, nonlinear least-squares fits were used to determine the values of $Q_{\max}$, *A*, and γ. Table 2a,b list the values of $Q_{\max}$ and $\gamma_{K}$, respectively, that result from fits at fixed values of *L* for each of the three networks. Similarly, Table 3a,b list the values of $Q_{\max}$ and $\gamma_{L}$ that result from fits at fixed values of *K* for each of the three networks.

The fitted values of $Q_{\max}$ increase systematically with increasing *L* and *K* and converge to statistically equivalent values at the largest ensemble sizes studied (320), regardless of whether *L* or *K* is increased. However, the values of $Q_{\max}$ are generally larger when fixing *L* rather than *K* when comparing results from when they are fixed at the same size. This fact implies that the maximum value of *Q* is approached faster by fixing *L* and increasing *K* rather than the opposite. The rate of convergence to $Q_{\max}$ is quantified by the γ exponents. The values of the exponents $\alpha_{K}\left(L\right)$ and $\alpha_{L}\left(K\right)$ depend on the network but only weakly vary with the values of *L* and *K*, respectively. For each network, however, the value of $\alpha_{K}$ is significantly larger than that of $\alpha_{L}$. Thus, the expected compute time increases faster with *K* than it does with *L*.

To understand these results, recognize that finding the best partition is an extremal process that, when repeated, is akin to the process of record-breaking. Let us recall some of the theory of the extreme value statistics of record-breaking [22]. Consider a sequence of independent and identically distributed random numbers(4)$$\left\{x_{1},x_{2},x_{3},\ldots ,x_{t},\ldots \right\}$$
chosen from a probability distribution of the form(5)$$p\left(x\right)=\mu B^{-\mu }{\left(B-x\right)}^{\mu -1},\;\;0\leq x\leq B,$$
where *B* is the maximum possible value of *x*, and define the *record*R(t) as the maximum value of *x* in the first *t* numbers in the sequence:(6)$$R\left(t\right)\equiv \max\left\{x_{1},x_{2},x_{3},\ldots ,x_{t}\right\}.$$

Then, in the limit of large *t*, the mean record will approach *B* as(7)$$\langle R\left(t\right)\rangle =B\left(1-\Gamma \left(1/\mu \right)\;t^{-1/\mu }\right),$$
i.e., as a power-law function with an exponent of 1/μ. From this, we see that μ=1 is a borderline case; from Equation (Equation 5), it is the case of a uniform distribution of *x*. If μ<1, then p(x) is maximal at x=B, and if μ>1, then p(x) vanishes as x→B.

While the analogy with this simple, analytically tractable model of record-breaking is not perfect, Equation (Equation 3) can be compared with Equation (Equation 7) by identifying $Q_{\max}$ with *B* and γ with 1/μ. Then, the fact that empirically $\gamma_{K}>1$ and $\gamma_{L}<1$ suggests that the distributions of the *Q* of the consensus partitions found by increasing *K* and *L* correspond to different cases. Namely, as *K* is increased, the consensus partition *Q* is likely to be near $Q_{\max}$ as it is for μ<1, while as *L* is increased, it is more likely to have a smaller value as it is for μ>1. To confirm this, we made 800 runs of RenEEL analyzing Network A with L=80 and K=320 and with L=320 and K=80. Figure 3 shows the consensus values of *Q* found in those runs. As expected, the values found with L=80 and K=320 (red bars) are much more likely to be near the maximum value than those found with L=320 and K=80 (blue bars).

We can now answer the central question of this paper: Given only limited computational resources, is it better to increase *K* or *L*? We have found that the average compute time grows faster with *K* than with *L*, but also that the consensus *Q* approaches $Q_{\max}$ faster with *K* than with *L*. Does the consensus *Q* approach $Q_{\max}$ as a function of compute time faster by increasing *K* or *L*? To answer this, we invert Equation (Equation 2) and combine it with Equation (Equation 3) to obtain(8)$$Q_{max}-Q∼{\langle t\rangle }^{-\gamma /\alpha }$$

So, the larger γ/α is, the faster *Q* approaches $Q_{max}$ as a function of average compute time. Table 4a shows the values of $\gamma_{K}/\alpha_{K}$ at different fixed *L* for the three networks. Similarly, Table 4b shows the values of $\gamma_{L}/\alpha_{L}$ at different fixed *L* for the three networks.

From these results, it can be clearly concluded that increasing *K* rather than *L* will cause *Q* to approach $Q_{max}$ faster. With limited computational resources, it is therefore better to increase *K* rather than *L*. Although we have shown that this is true only for three example networks and only when maximizing Modularity, we speculate that these networks are not special and maximizing modularity, rather than a different objective function, is also not special. Therefore, the conclusion that it is more computationally efficient to increase *K* rather than *L* in RenEEL should generally be true. However, it would be interesting to explore this question when maximizing other objective functions with RenEEL.

## Figures and Tables

**Figure 1 entropy-27-00628-f001:**
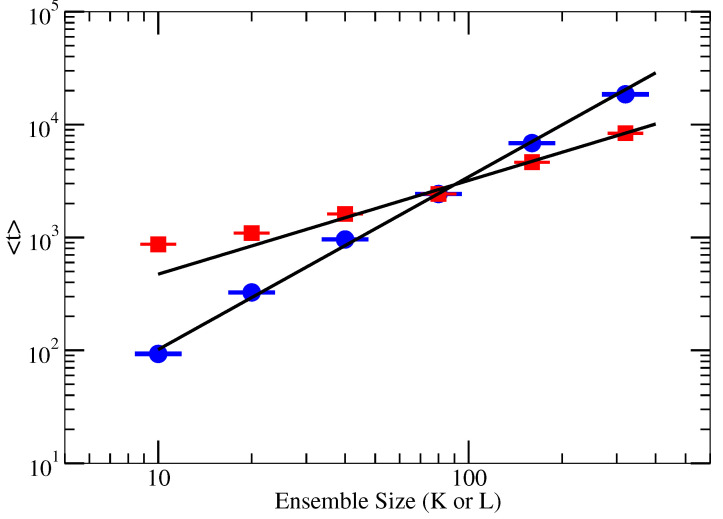
The compute time for RenEEL to complete when analyzing Network A. Data are measured in seconds. Blue circles are results as a function of *K* for fixed L=80, and red squares are results as a function of *L* for fixed K=80. The straight lines are power-law fits to the data of ensemble sizes greater than 10. The slope of the fit to the blue circles is $\alpha_{K}=1.489\left(4\right)$, and to the red squares is $\alpha_{L}=0.810\left(3\right)$.

**Figure 2 entropy-27-00628-f002:**
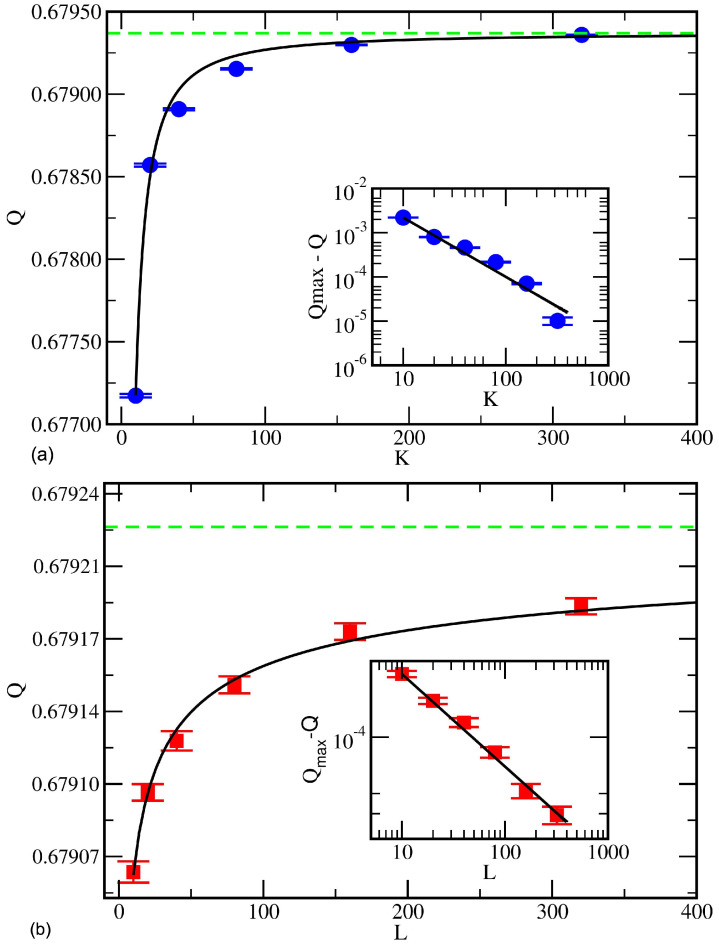
Modularity *Q* of RenEEL consensus partitions for Network A (**a**) as a function of *K* for fixed L=80 (blue circles), and (**b**) as a function of *L* for fixed K=80 (red squares). The black solid lines are fits to the data, representing a power-law approach to the estimated maximum value $Q_{\max}$ indicated by the green dashed lines. The value of $Q_{\max}$ is 0.679368(6) in (**a**) and 0.679229(40) in (**b**). The insets show the data on log-log axes. The slope of the fit to the blue circles in the inset of (**a**) is $\gamma_{K}=1.34\left(2\right)$ and to the red squares in the inset of (**b**) is $\gamma_{L}=0.36\left(18\right)$.

**Figure 3 entropy-27-00628-f003:**
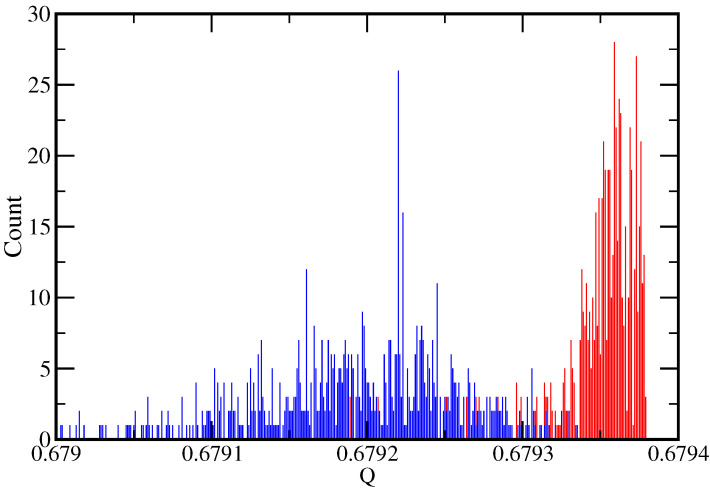
Distribution of *Q* obtained for Network A. Blue bars correspond to the data for L=320 and K=80, and red bars to L=80 and K=320.

**Table 1 entropy-27-00628-t001:** (a) Values of scaling exponent $\alpha_{K}$ of the compute time divergence at fixed values of *L*. (b) Values of scaling exponent $\alpha_{L}$ of the compute time divergence at fixed values of *K*.

(**a**)			
L	**Network A**	**Network B**	**Network C**
20	1.382(2)	1.262(2)	1.331(4)
40	1.431(2)	1.212(2)	1.300(4)
80	1.489(4)	1.240(2)	1.300(2)
160	1.483(2)	1.164(2)	1.320(4)
320	1.469(2)	1.194(4)	1.311(6)
(**b**)			
K	**Network A**	**Network B**	**Network C**
20	0.732(3)	0.800(6)	0.821(4)
40	0.752(2)	0.820(4)	0.872(4)
80	0.810(3)	0.851(2)	0.900(8)
160	0.810(4)	0.800(2)	0.924(4)
320	0.792(3)	0.879(4)	0.898(6)

**Table 2 entropy-27-00628-t002:** (a) Values of asymptotic maximum Modularity $Q_{max}$ found at fixed values of *L*. (b) Values of scaling exponent $\gamma_{K}$ of the Modularity convergence to $Q_{\max}$ at fixed values of *L*.

(**a**)			
L	**Network A**	**Network B**	**Network C**
10	0.679265(4)	0.886761(2)	0.742303(12)
20	0.679309(6)	0.886784(6)	0.742589(18)
40	0.679320(6)	0.886814(2)	0.742829(16)
80	0.679368(6)	0.886825(4)	0.742956(18)
160	0.679370(8)	0.886832(2)	0.743013(16)
320	0.679377(4)	0.886834(2)	0.743168(20)
(**b**)			
L	**Network A**	**Network B**	**Network C**
10	1.21(2)	2.16(4)	1.18(18)
20	1.29(2)	2.15(6)	1.31(20)
40	1.38(2)	2.01(8)	1.28(20)
80	1.34(2)	2.04(4)	1.71(16)
160	1.35(4)	2.29(14)	1.22(18)
320	1.25(2)	2.05(6)	1.29(20)

**Table 3 entropy-27-00628-t003:** (a) Values of asymptotic maximum Modularity $Q_{max}$ found at fixed values of *K*. (b) Values of scaling exponent $\gamma_{L}$ of the Modularity convergence to $Q_{\max}$ at fixed values of *K*.

(**a**)			
K	**Network A**	**Network B**	**Network C**
10	0.677480(34)	0.886521(12)	0.740259(170)
20	0.678725(24)	0.886763(10)	0.741572(174)
40	0.678963(36)	0.886830(12)	0.743070(172)
80	0.679229(40)	0.886832(10)	0.743072(170)
160	0.679246(26)	0.886832(6)	0.743077(172)
320	0.679370(20)	0.886833(8)	0.743088(168)
(**b**)			
** K **	**Network A**	**Network B**	**Network C**
10	0.37(6)	0.86(10)	0.29(26)
20	0.49(8)	0.88(10)	0.29(22)
40	0.50(12)	0.85(16)	0.22(24)
80	0.36(18)	0.85(12)	0.20(22)
160	0.47(20)	0.69(12)	0.25(20)
320	0.67(10)	0.86(16)	0.25(20)

**Table 4 entropy-27-00628-t004:** (a) Values of scaling exponent $\frac{\gamma }{\alpha }$ at fixed values of *L*. (b) Values of scaling exponent $\frac{\gamma }{\alpha }$ at fixed values of *K*.

(**a**)			
L	**Network A**	**Network B**	**Network C**
20	0.93(1)	1.70(5)	0.98(15)
40	0.96(1)	1.65(6)	0.98(15)
80	0.90(1)	1.64(3)	1.31(12)
160	0.91(2)	1.96(12)	0.93(13)
320	0.89(1)	1.71(5)	0.98(12)
(**b**)			
K	**Network A**	**Network B**	**Network C**
20	0.66(11)	1.09(13)	0.35(26)
40	0.66(16)	1.03(20)	0.25(26)
80	0.44(22)	0.99(14)	0.22(24)
160	0.57(22)	0.88(15)	0.27(21)
320	0.68(12)	0.97(18)	0.28(22)

## Data Availability

The three networks used in this study are openly available at https://sites.cc.gatech.edu/dimacs10/archive/clustering.shtml (accessed on 19 April 2023). The raw data, which are used for analysis, are provided in Appendix A.

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
