# Peer review of "Extreme Value Statistics of Community Detection in Complex Networks with Reduced Network Extremal Ensemble Learning (RenEEL)"

_entropy, 2025, doi:10.3390/e27060628_

Round 1
Reviewer 1 Report
Comments and Suggestions for Authors
1. The aim of the paper isn't clearly specified .
2. The authors didn't clearly mention how they come up with the choices of L and K for the networks investigated.
3. The results does not provide convincing evidence regarding the claim made.
4. Why the networks investigated should be considered as representative? What if the networks are way larger than the ones considered? What if there are connected components in the network?
5. Tables show that increasing K and L contribute to the increment of modularity. It is not clear which one is an effective way?
6. Authors didn't clearly mention how they find the consensus partition.
Reviewer 2 Report
Comments and Suggestions for Authors
In this paper, the authors explore a machine-learning algorithm RenEEL, that refines an ensemble of K partitions by iteratively replacing the weakest partition with the best of L partitions found by analyzing a simplified network. This process continues until the ensemble reaches consensus. The authors analyze three real-world networks, and find that increasing K (ensemble size) is generally more effective than increasing L (simplified partitions) for achieving best partitions, also with limited computational resources. The topic of the research is very interesting and is within the academic agenda. Overall, the paper is scientifically rigorous, well-written, and presents results that highlight interesting properties of the proposed approach. Nevertheless, I have several observations and concerns that I believe should be addressed to strengthen the manuscript. These are outlined below:
-
My first comment goes with respect to the use of Modularity method. This method categorically fails in its own stated goal, since it always finds high-scoring partitions in networks sampled from its own null model. Please, have a look at references by Tiago P. Peixoto who systematically showed and discussed the applications of this method to community detection [https://skewed.de/lab/research.html; Peixoto T.P. Descriptive vs. inferential community detection in networks: Pitfalls, myths and half-truths Elements in the Structure and Dynamics of Complex Networks, Cambridge University Press (2023); Guimerà R., Sales-Pardo M., Amaral L.A.N. Modularity from fluctuations in random graphs and complex networks, Phys. Rev. E, 70 (2004), Article 025101].
-
My other concern goes to Networks types considered in the study. Have you considered undirected and unweighted networks only? What about other network types? For instance financial and economic networks. Directed and weighted networks introduce additional complexity and implications, can alter the Modularity calculation and may require adapting the RenEEL algorithm to handle the added dimensions. Please clarify in your text this point.
-
How do you handle dynamic and temporal networks? I understand that the present study is only designed for static networks, since you are using as a base algorithm a Modularity method, which assumes a fixed network structure, and can not be directly applied to dynamic or temporal networks. This needs to be clarified in the manuscript. The authors may also include a discussion on the limitations of their approach for dynamic and temporal networks and suggest how their approach can be adapted to these context.
-
Is the main result “increasing K is generally more effective than increasing L for finding the best partition” a novelty? As K increases, each community becomes smaller, effectively reducing the computational cost per partition. In turn, small communities are easier to identify. But also, this process leads to splitting larger meaningful communities into smaller less important ones. When K becomes too large, one may have overfitting and obtain bad communities. I assume that RenEEL helps to avoid this but it would be helpful if authors shed light on it. Also please highlight if the obtained result is a new result in this area or confirmation of a known result.
-
What about determining the optimal value of K?
-
What about the balance of scalability and interpretability? This has to do with the optimal value of K, and avoiding over segmentation and the loss of meaningful community structure. How do the authors evaluate and ensure the quality of the results in terms of communities obtained? This point is important especially for those who apply community detection in real world networks and are interested in the composition of clusters. Please, clarify this point.
Round 2
Reviewer 1 Report
Comments and Suggestions for Authors
Authors have provided clarifications/modifications to address the comments from the first round of reviews.
Reviewer 2 Report
Comments and Suggestions for Authors
I am pleased to see that the authors engaged constructively with the reviewers' feedback and have addressed a number of the concerns and suggestions. These efforts have resulted in a better version of the manuscript. I particularly appreciate the authors' detailed and thoughtful responses to some of my comments. To clarify, the comments in my earlier report were intended as constructive suggestions aimed to enhance the clarity and accessibility of the paper. They were not meant to imply that the authors should "solve all the problems in community detection", a task that, as the authors themselves note, would be neither feasible nor expected in a single contribution. My primary concern was that the initial submission lacked several key methodological and contextual clarifications necessary for readers to fully appreciate the contribution, applicability, and relevance of the work. It is not ideal for readers to rely on previous publications by the authors to fill in these gaps.
The revised manuscript represents a significant improvement, and meets the standards for publication in its current form. I therefore recommend that the paper be accepted.